# Assessing the medium-term impact of a home-visiting programme on child maltreatment in England: protocol for a routine data linkage study

Fiona V Lugg-Widger,[1] Rebecca Cannings-John,[1] Sue Channon,[1]
Deborah Fitzsimmons,[2] Kerenza Hood,[1] Kerina H Jones,[3] Alison Kemp,[4]
Joyce Kenkre,[5] Mirella Longo,[6] Kirsten McEwan,[7] Gwenllian Moody,[1]
Eleri Owen-Jones,[1] Julia Sanders,[8] Jeremy Segrott,[1,9] Michael Robling[1]

For numbered affiliations see end of article.

**Correspondence to**
Fiona V Lugg-Widger; LuggFV@cardiff.ac.uk

## ABSTRACT

**Introduction** Child maltreatment involves acts of omission (neglect) or commission (abuse) often by caregivers that results in potential or actual harm to a child. The Building Blocks trial (ISRCTN23019866) assessed the short-term impact of an intensive programme of antenatal and postnatal visiting by specially trained nurses to support young pregnant women in England. The Building Blocks: 2–6 Study will assess the medium-term impacts of the programme for mothers and children (n=1562), through the linkage of routinely collected data to the trial data, with a particular emphasis on the programme's impact on preventing child maltreatment.

**Methods and analysis** We have developed a bespoke model of data linkage whereby outcome data for the trial cohort will be retrieved by linked anonymous data abstraction from NHS Digital, Office for National Statistics and the Department for Education's National Pupil Database. Participants will be given reasonable opportunity to opt out of this study prior to data transfer. The information centres will match participants to the information held in their databases using standard identifiers and send extracts to a third-party safe haven. The study will have 80% power to detect a 4% difference (4%vs8%) for the binary primary outcome of child in need status (from birth to key stage 1) at a two-sided 5% alpha level by following up 602 children in each trial arm. Analysis will be by intention to treat using logistic multilevel modelling. A cost-and-consequences analysis will extend the time frame of the economic analysis from the original trial.

**Ethics and dissemination** The study protocol has been approved by the National Health Service Wales Research Ethics Committee and the Health Research Authority's Confidentiality Advisory Group. Methods of innovative study design and findings will be disseminated through peer-reviewed journals and conferences; results will be of interest to clinical and policy stakeholders in the UK.

**Trial registration number** ISRCTN23019866.

## Strengths and limitations of this study

► This study aims to provide much-needed evidence about the medium-term benefits of the Family Nurse Partnership programme in England. This study has the capacity to either confirm the current perspective on the value of the intervention or demonstrate clinically meaningful benefits to children in vulnerable young families

► There are distinct benefits associated with using routine data including a reduction in cost and participant burden over prospectively collected data and relative completeness and therefore minimisation of bias over self-report, particularly for such sensitive outcomes

► The establishment of a regulatory secure research database for this cohort of trial participants also offers the prospect of further data being added over the longer term and of broadening the scope of the dataset to other outcome domains relevant to this intervention, such as criminal justice and welfare benefits

► The extent of this benefit will be balanced by our ability to adequately access the data from information centres in a timely fashion, the quality of matching conducted as well as the quality of the data ultimately retrieved

## INTRODUCTION
### Maltreatment
Child maltreatment involves acts of omission (neglect) or commission (abuse) often by caregivers who inflict harm or fail to act to prevent harm to a child.[1] Abuse may be physical, emotional or sexual. Neglect represents persistent failure to meet basic physical or psychological needs, often resulting in serious impairment of the child's health and/or development.[1] Neglect may involve failing to protect a child from physical and emotional harm or danger, provide adequate supervision

or ensure access to appropriate medical care. In the year ending 31 March 2015 in England, there were 635 600 referrals to children's social care (CSC) services, 403 400 children starting an episode of need (an overall rate of 348.0 per 10 000) and 62 200 children became subject of a child protection plan.[2] Of children who became subject of a child protection plan, the most common initial category of abuse was neglect (43.2%) followed by emotional abuse (33.7%).

In the UK, preventing maltreatment is an important focus of government concern. The Children Act 1989 specifies agencies' responsibilities to cooperate in the interests of vulnerable children, for children in need (section 17) and children suffering or likely to suffer from significant harm (section 47). A child in need (CIN) is defined as a child who is unlikely to achieve or maintain a reasonable level of health or development or whose health and development is likely to be significantly or further impaired, without the provision of services or is a child who is disabled. Local authority provisions may include supervision of activities, financial help, provision of family accommodation, respite or home help in addition to advice and guidance from social workers.

### Family Nurse Partnership home-visiting programme

There has been increasing emphasis on the primary prevention of child maltreatment, including interventions directed at general populations and those targeting high-risk groups.[3] One such intervention is the Family Nurse Partnership (FNP) programme (developed in the USA as the Nurse Family Partnership (NFP))—a home-visiting approach with three overarching goals: to improve birth outcomes, optimise child health and development including reducing maltreatment, and promote economic self-sufficiency of mothers.[4]

In three US trials (in Elmira, Memphis and Denver),[5–7] the NFP has demonstrated improvements in prenatal health behaviours and birth outcomes, sensitive child care, maternal life course (eg, greater workforce participation) and child and adolescent functioning. It has also shown positive effects in relation to reductions in rates of child injuries, abuse and neglect. In the first US trial in 1977, a subgroup analysis of poor unmarried teens (54 families) revealed that, by age 2, there was verified abuse/neglect in 19% of control children compared with 4% in the group in receipt of NFP in both pregnancy and infancy (treatment difference of 0.15, 95% CI of −0.01 to 0.31) and 56% relative reduction in emergency department encounters for injuries and ingestions during the second year of life.[5] Among the subgroup of children (56 families) with a state-verified report of maltreatment by age 4, the NFP group of children exhibited fewer risks for harm than the control group (eg, fewer attendances with injuries/ingestions, safer home environment) at follow-up points between 25 and 50 months of life.[8] This was considered to be due to the earlier and more comprehensive detection of maltreatment by nurse-visited families.

The NFP programme was adapted for implementation as the FNP and was introduced in England in 2007. Our Building Blocks trial (ISRCTN23019866) was the first trial of FNP in England and evaluated short-term outcomes to age 2—the duration of the FNP programme.[9] The trial reported no difference for four primary outcomes: biomarker-calibrated self-reported tobacco use by the mother at late pregnancy, birth weight of the baby, the proportion of women with a second pregnancy within 24 months postpartum and emergency attendances and hospital admissions for the child within 24 months postpartum.[10] We observed some differences for secondary child development outcomes including the rate of safeguarding events reported in primary care records. While the current evidence does not support continuation of the programme in England, previous evaluations have demonstrated benefit over the longer term (eg, up to 15 years of age).[11] For maltreatment outcomes, this benefit has been increasingly evident after the age of 4 years;[12] therefore, the current study will establish whether FNP has moderated maltreatment outcomes over a medium-term period of follow-up (ie, to the point where the child is aged 6 years old).

## METHODS AND ANALYSIS
### Research objective

The Building Blocks: 2–6 Study (BB:2-6) will use data linkage of routinely collected national datasets to assess the medium-term impact of the FNP intervention on child maltreatment outcomes and key indicators of neglect.

### Study design

This is a data linkage study, which will generate a linked anonymised database hosted by an independent trusted third party. Participant mothers and children from Building Blocks: 0–2 (BB:0–2) will be followed up for a further 4 years using routine data only. Data from various routine public sector sources will be retrieved and linked to the trial data to enable children and mothers to be followed until the child reaches key stage 1 (the 2 years of schooling when pupils are aged between 5 and 7). The study formally started in February 2014 and will report to the funder in May 2018. Participants were recruited to the trial between June 2009 and July 2010 and the 6-year follow-up ends (ie, the last child will have turned 6) in March 2017. A summary of the data sources is provided in table 1, and the time period for each dataset is shown in figure 1 . Study outcomes are summarised in table 2.

### Data providers and datasets
#### The BB:0–2 trial data

Data collected for the initial trial will be used in the present study.[9 10] A baseline home assessment was conducted on trial entry using computer-assisted personal interview (CAPI). Follow-up was by computer-assisted telephone interview at 34–36 weeks gestation and 6, 12 and 18 months postnatal. A final home-based CAPI was conducted at 2 years after birth. Several routinely collected datasets were

**Table 1** Summary of data sources

| BB:0–2 | BB:2–6 | Provided by | Dataset | Time period* | Eligibility/coverage | Mother | Child | Indicative/key data items |
|---|---|---|---|---|---|---|---|---|
| ✓ | | Trial participants' maternal self-report | Baseline | 2009–2013 | Trial participants | Yes | No | Socioeconomic; maternal health and well-being; health behaviour; pregnancy complications, neonatal outcomes; feeding and development |
| ✓ | | | Late pregnancy | | | | | |
| ✓ | | | 6 months | | Postbirth | | Yes | |
| ✓ | | | 12 months | | | | | |
| ✓ | | | 18 months | | | | | |
| ✓ | | | 24 months | | | | | |
| ✓ | | Maternity records | Maternal outcomes | 2009–2010 | UK | Yes | Yes | Maternal health and well-being; neonatal outcomes |
| ✓ | | GP records | GP consultations | 2009–2013 | UK | Yes | Yes | Immunisations; safeguarding |
| ✓ | | PCTs | Immunisation | 2009–2013 | England | No | Yes | Immunisations |
| ✓ | ✓ | DoH | Abortions | 2009–2013 | England and Wales; all abortions performed in the NHS or an approved independent sector | Yes | No | Abortions |
| ✓ | ✓ | ONS | Mortality records | 2009–2017 | UK | Yes | Yes | Mortality data |
| ✓ | ✓ | NHS Digital/HES | Inpatient | 2009–2017 | Any NHS hospital in England | Yes | Yes | Injuries and ingestions; subsequent pregnancies |
| ✓ | ✓ | | Outpatient | | | | | |
| ✓ | ✓ | | A&E | | | | | |
| | ✓ | Department for Education/NPD | CIN | 2009–2017 | Registered with social services in England | <18 years | Yes | Yes | CIN status and CLA status |
| | ✓ | | CLA | | | | | |
| | ✓ | | EYFSP | 2013–2017 | Public schools in England | 4 years | No | Yes | Indicators of maltreatment; educational development and attainment; eligible for free school meals |
| | ✓ | | Census | | | 2–19 years | | | |
| | ✓ | | Alt Provision | | | 2–19 years | | | |
| | ✓ | | PRU | | | 2–19 years | | | |
| | ✓ | | Key stage 1 | 2016–2017 | | 5–7 years | No | Yes | |

*Trial started 2009; 2-year follow-up ended in 2013; 6-year follow-up ends in 2017.
A&E, accident and emergency; CIN, child in need; CLA, child looked after; DoH, Department of Health; EYFSP, Early Years Foundation Stage Profile; GP, general practitioner; HES, Hospital Episode Statistics; NPD, National Pupil Database; ONS, Office for National Statistics; PCT, Primary Care Trust; PRU, pupil referral unit.

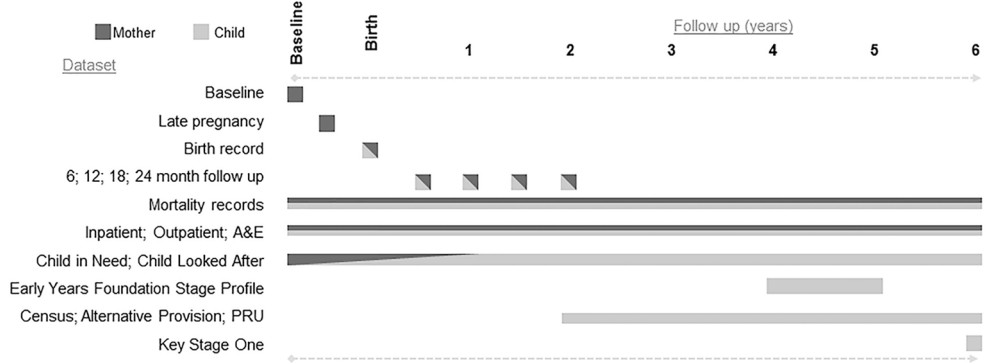

**Figure 1** Follow-up and datasets over the six years. A&E, accident and emergency; PRU, pupil referral unit.

accessed, and data were obtained from the following sources: maternity records (medical and obstetric history items, antenatal attendances and maternal and neonatal outcomes), primary care notes for each mother and child dyad (consultations, immunisations, pregnancies, safeguarding), abortions data from the Department of Health (DoH) abortions statistics team and immunisation data via Cover of Vaccination Evaluated Rapidly contacts.

### NHS Digital

The Hospital Episode Statistics (HES) datasets hold records on over 125 million hospital admissions, outpatient and accident and emergency episodes each year. Data can be requested from NHS Digital (formerly known as the Health and Social Care Information Centre), the executive non-departmental public body established under the Health and Social Care Act 2012.[13] All available records belonging to cohort members (mothers and children) will be obtained from study entry of the mother, which occurred between

| Table 2 Study outcomes | | | | |
|---|---|---|---|---|
| **Domains** | **Outcomes** | **HES** | **ONS** | **NPD** |
| *Primary:* | | | | |
| CIN status recorded at any time during the follow-up period | CIN status as of 31 March each year | | | ✓ |
| *Secondary:* | | | | |
| (1) Objective measures of maltreatment | Child protection registration | | | ✓ |
| | Details of a child protection plan | | | ✓ |
| | CIN categorisation | | | ✓ |
| | CIN duration | | | ✓ |
| | Looked after status | | | ✓ |
| | CLA period of care | | | ✓ |
| | Legal status of CLA | | | ✓ |
| | Cause of death | | ✓ | |
| (2) Associated measures of maltreatment | DNA appointments | ✓ | | |
| | Injuries and ingestions | ✓ | | |
| (3) Intermediate FNP programme outcomes | Subsequent pregnancies | ✓ | | |
| (4) Costs | Health and social care resource use | ✓ | | ✓ |
| (5) Child health, developmental and educational outcomes | Special educational needs | | | ✓ |
| | Disability | ✓ | | ✓ |
| | Day care attendance | | | ✓ |
| | Early-years assessment | | | ✓ |
| | School attendance | | | ✓ |
| | Key stage 1 attainment | | | ✓ |

CIN, child in need; CLA, child looked after; DNA, did not attend; FNP, Family Nurse Partnership; HES, Hospital Episode Statistics; NPD, National Pupil Database; ONS, Office for National Statistics.

June 2009 and July 2010 until the date the child turns 6. The data requested include diagnoses, procedures, length of episode and external causes of injuries coded according to the 10th revision of the International Statistical Classification of Diseases and Related Health Problems (ICD-10) codes.[14]

NHS Digital has responsibility for collecting these data from across the health and social care system to allow NHS hospitals to be paid for the care they deliver. At the end of the financial year (March), a final dataset is collated. This dataset is cleaned and validated before being available for research at the end of the year (December).

### Office for National Statistics

The Office for National Statistics (ONS) collects information on cause of death from civil registration records. Mortality data can be accessed through NHS Digital. For registered deaths, the underlying cause of death is derived from the sequence of conditions leading directly to the death and is recorded on the death certificate. Deaths are subsequently coded in line with the ICD-10.

### Department for Education

The Department for Education (DfE) holds information on pupils throughout the different phases of education. Records are sourced from publicly funded schools, local authorities and awarding bodies and held in the National Pupil Database (NPD). Datasets are available on various aspects of education such as School Census data, absence data and school attainment.[15] All available records for the children in the cohort will be obtained from the various datasets held. Data coverage will vary depending on the dataset in question. For example, the School Census returns data on maintained schools (funding and oversight is through the local authority), which represents the majority of schools, academies (funding and oversight is from DfE), city technology colleges, maintained and non-maintained special schools and hospital special schools. Schools that are entirely privately funded and home education are not included in the data; this represents 7% of English students.[16]

In the UK, education is mandatory from the first school term after their fifth birthday. Prior to this, some children may not have received formal early-years provision and therefore may not appear in the datasets. A survey conducted in 2014–2015 commissioned by DfE reported that 25% of children aged 0–4 were not in receipt of any early-years provision. Older preschool children (aged 3–4), however, were far more likely to receive early-years provision (92%) than younger preschool children (aged 0–2) (61%).[17] We would therefore expect similar coverage rates for this study.

The data requested include the number of hours attended, early educational development, eligibility for free school meals and special educational needs provision type. Datasets are collated throughout the year and are available at set time points annually.

### Social care data

Social care data from local authorities are available through the NPD via two datasets, CIN and child looked after (CLA). The CIN census captures individual level information on children referred to and assessed by children's social care services within each 12-month period.[18] CLA is collected in the SSDA903 return, an annual statutory data collection for all local authorities.[19] Any child in the cohort who is in one of these datasets will be identified. Mothers who were <18 years at the time of participation in the BB:0–2 trial will also be identified in these datasets. There will not be the coverage issues as seen in the education data returns, and importantly, the primary outcome will be sourced from these social care datasets.

### Study participants: inclusion and exclusion criteria

Eligible participants are those mothers recruited to the BB:0–2 trial and their first child (or twins, if relevant) and who were not mandatorily withdrawn from the study or electively withdrew including their consent for use of their data. Women were recruited as nulliparous women aged 19 or under, living in one of 18 local authority FNP catchment areas; recruited by 24+6 weeks gestation, have conversational level of English and were able to consent to research.[10]

Children in medium-term foster placements or adopted within the 6-year study period can be linked up to the date of adoption. Maternal or child death will be captured as an outcome.

### Recruitment/dissent

Participants previously consented to enter into the BB:0–2 trial and provide self-report and access to their routine records for the period up to 2 years post partum. In order to obtain an unbiased estimate of the medium-term effect of FNP on objective and associated maltreatment outcomes, we have received section 251 (s251) support of the 2006 NHS Act approval from the Health Research Authority's Confidentiality Advisory Group (HRA CAG) to pass identifiable participant data legally held by Cardiff University to the information centres (ICs) to link to routine data. This is without obtaining further consent from participants, instead using an opt-out/dissent model.

### Justification of approach

Consent for longer-term follow-up (ie, beyond 24 months post partum) was originally proposed in the BB:0–2 trial. However, on ethical review, it was considered that greater specificity about exact outcomes than could be provided at recruitment was required. Additionally, providing meaningful consent for much longer follow-up was also challenging, particularly on behalf of yet-to-be-born children.

Developing the opt-out approach was necessary due to (1) the child protection focus of the study and the consequent sensitivity and impracticality in asking directly for consent, (2) the mobility and relative difficulty in

ongoing direct access to these participants, (3) the consequent introduction of non-ascertainment bias on sample representativeness—resulting in a non-random sample—and (4) the likely cost and logistical requirements of securing even modest levels of additional consent.

## Methods of notifying participants

We discussed the issue of dissent and fair processing with the HRA CAG and have subsequently attempted to contact all mothers recruited to the original BB:0–2 trial to inform them that medium-term follow-up using anonymised records will be undertaken.

Details of participants' residential addresses were updated using their most recent address registered with their general practitioner. Where available, mobile number and email addresses collected for the trial were used to send SMS and emails to participants. All three modes of contact were used over a 2-day period, and participants were provided with a 2-month window in which to contact the project team to discuss the project and opt-out if they wished. A website was also available with the same information, which directed participants to contact the project team if they wished.

### Development of opt-out letter

A group of care-experienced young people (CASCADE Voices)[20] advised on the layout, wording and tone of a letter to be sent to all participants. A key consideration was to communicate the focus of this follow-on study in a sensitive manner. The final letter was approved by both an NHS Research Ethics Committee and CAG committee as part of overall governance approval for the study. The letter contained information on the trial, the follow-on study and a flowchart for what to do if women wished to discuss the project and/or opt out.

### Process to manage dissent

Women notifying the study team of their dissent will be recorded as 'opted out,' removed from all project datasets for this follow-up work, and identifiable datasets are to be sent to ICs. They will not be included in any of the datasets or analyses for this follow-on study.

## Governance and compliance

Following ethical approval (14/WA10062) and s251 support (CAG 10-08(b)/2014), data request applications were submitted to DfE, NHS Digital and ONS.

In order to satisfy the requirements of the s251 support and NHS Digital contract, the Information Governance (IG) Toolkit self-assessment[21] (commissioned by the Department for Health for NHS Digital to develop and maintain) was required. This organisation-level assessment provides reassurance of satisfactory information governance within the host trials unit. Both the s251 support and IG Toolkit are assessed and renewed on an annual basis. The opt-out model was also required to satisfy s251 support as well as the DfE assessment of compliance with principle 1 of the Data Protection Act 1998. Governance and IC requirements prior to application approval are shown in figure 2 .

## Data matching

Maternal and child identifiers will be sent to both NHS Digital and DfE for matching with their databases. Each IC holds differing identifiers including a unique identifier for each individual (NHS number; unique pupil number (UPN)).

Matching with HES data will be by exact matching on NHS number, date of birth, postcode and gender. This was conducted for BB:0–2 and achieved a high match rate where 99.6% of mothers' and babies' records were matched fully (ie, matched on all identifiers provided)

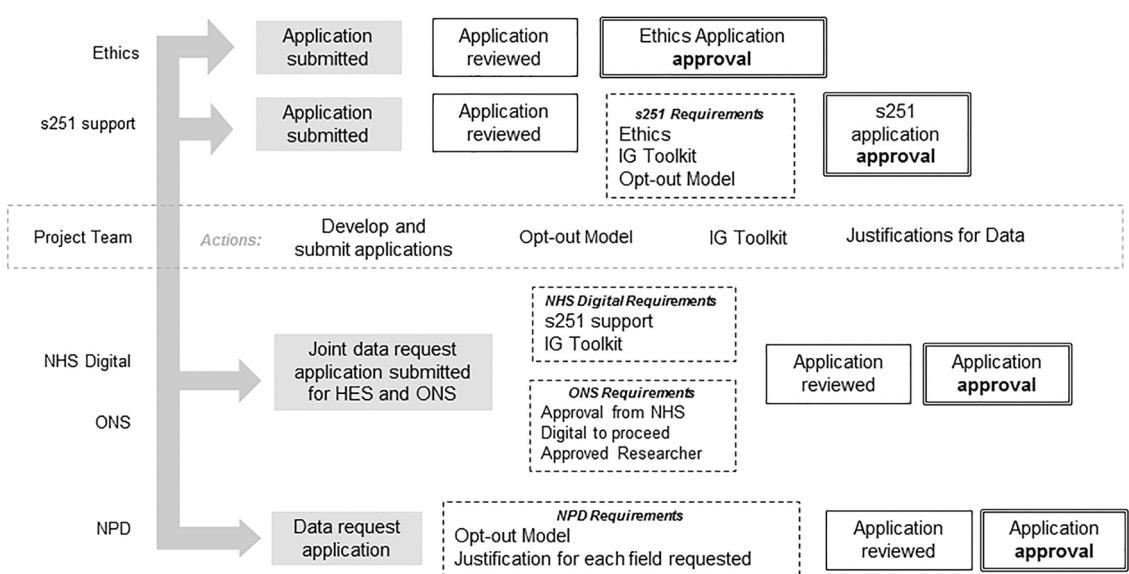

**Figure 2** Governance and information centre requirements prior to application approval. s251, section 251 of the NHS 2006 Act; ONS, Office for National Statistics; NPD, National Pupil Database; IG Information Governance.

or partially (ie, matched on a reduced but acceptable number of identifiers provided). This will be repeated for this study. NHS Digital will then exact match with ONS using NHS number in order to obtain mortality data.

As NPD does not include NHS numbers, initially, exact matching on first name and surname, date of birth and postcode (of both mother and child for social care data; all other datasets, just child) will be undertaken. Further matching required will be by fuzzy matching of first name. The CIN and CLA datasets do not contain names or post-codes. Therefore the matching will be in two phases: (1) participants will be matched with NPD, and UPN will be added to all participants and (2) this will be used to identify individuals in the CIN and CLA datasets.

Data matching at DfE and NHS Digital/ONS are independent; therefore, match rates at the participant level are expected to vary (some may match to NPD but not HES). Educational records should be available for all children in the trial cohort, whereas health and social care derived data will only exist where the child has received a relevant episode of care. Participants will be compared using trial baseline data to check for any bias between those who are matched and not matched for those datasets where they would all be expected to be present (eg, School Census for all children).

### Pseudonymised dataset

A unique study ID will be attached to each participant's record prior to data transfer to ICs. Once ICs have matched records to their database, only the unique study ID is retained. Data extracts from both ICs plus data files from the trial (following a process of de-identification and standardisation in Cardiff to reduce risk of later unintentional participant level identification) will all be securely transferred to a data safe haven,[22] the Secure Anonymised Information Linkage (SAIL) databank, for linking and storage. The data flow is shown in figure 3.

A SAIL data analyst will reassign the study ID with a new anonymous linking field (ALF) and store the corresponding ID in a separate encrypted password-protected file.[23]

Participants will not be identifiable to the study team or to the SAIL analyst, but incoming datasets can be linked at the individual level using the ALF. The study team will have controlled remote access to these data, thus ensuring the security of the pseudonymised database.[24] All data cleaning and analysis will be carried out via the remote portal by the study data manager and statistician.

Data from NHS Digital and NPD will be requested at two time points. The first data extract will confirm the data flow model and assess data quality and the suitability of data for answering key study analyses. The second data request will be made once all children in the study have reached key stage 1 (April 2017) and on which the study findings will be reported on (2018).

### Control of data

Cardiff University controls under contract the identifiable trial data that are being transferred to the ICs and to the safe haven. Data held by NHS Digital, ONS and DfE, for which they are the controllers, are de-identified and then sent to SAIL to be linked and held (including

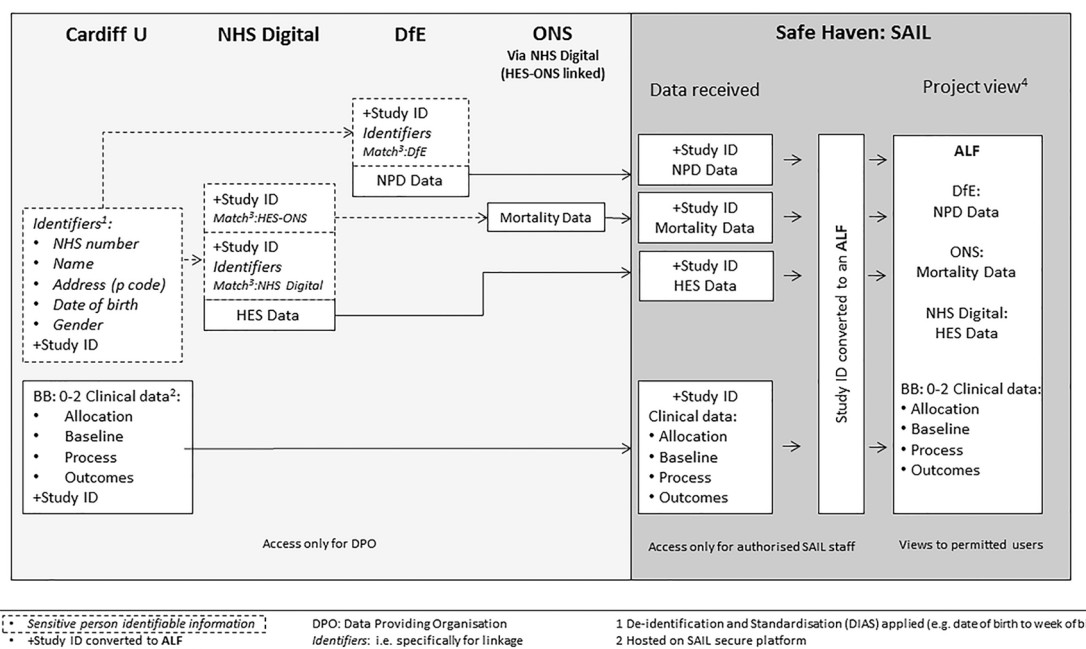

**Figure 3** Data flow. (1) Participant identifiable information securely transferred for linkage. (2) De-identification and standardisation applied (eg, date of birth to week of birth). (3) Information centres confirm matching of participant identifiers. (4) Hosted on SAIL secure platform. ALF, anonymous linking field; BB:0–2, the Building Blocks trial; DfE, Department for Education; DPO, data-providing organisation; HES, Hospital Episode Statistics; ONS, Office for National Statistics; SAIL, Secure Anonymised Information Linkage.

the de-identified trial data) in a secure anonymised standalone database for use by nominated study team members. SAIL will control the safe haven environment and will process the pseudonymised data for secure use by study team. Cardiff University will control the purposes to which the data are put in answering research questions as per the study protocol. Once linked in the data safe haven, the ability to submit queries to each IC about individual records will be more limited than if identifiable data were returned to the research team in Cardiff. Data cleaning will remain possible, however, as will generic queries about data provided in batch. The quality of matching conducted by NPD and NHS Digital/ONS will be a key factor in the success of the study.

## Analysis

### Power calculation

Primary outcome (CIN status at any point between birth and 6 years): for CIN status, available UK data on rates are not specific to the age range of interest, but the rate in the general population aged 5–9 years is 4.6% (for local authorities comprising study sites in BB:0–2). The rate of CIN status would be expected to be greater in the specific study sample, and therefore, we have assumed a rate of 8%. We hypothesise that FNP would reduce the detection of CIN in the first 6 years and thus assumed a difference of 4% as being important. To detect a difference of 4% (FNP of 4% vs usual care of 8%) would require 602 children in each arm (1204 in total) using 80% power and a two-sided 5% alpha level.

BB:0–2 recruited 1645 women, with 1562 available for follow-up (ie, excluding those subject to a mandatory withdrawal). Follow-up through medical records (assuming 10% loss in tracking and linkage) would result in 1405 participants, thus securing enough data to test the primary outcome.

### Main analysis

Analyses will be conducted on an intention-to-treat basis and due emphasis placed on CIs for the between-arm comparisons. Descriptive statistics of demographic and baseline measures will be used to ascertain any marked imbalance between the trial arms. The primary comparative analysis on CIN status at any point between birth and 6 years will use logistic multilevel modelling to investigate differences between the groups, and odds ratios alongside 95% CIs will be reported. Multilevel modelling will allow for clustering of effect within a site and family nurse. Modelling the impact of key subgroups (deprivation; looked after status of mother; adaptive functioning; not in education, employment or training status; and age) and different intervention elements (eg, gestational age at programme entry, dosage) on outcome will be undertaken by extending the primary models and testing for interaction effects. The role of potential moderators of programme effect (eg, domestic violence) will also be explored.

Secondary outcomes will assess group differences in objective and associated measures of maltreatment, intermediate FNP programme outcomes as well as child health, development and educational outcomes (as detailed in table 2). The majority of these are binary outcomes (presence/absence of a status, meeting the key stage 1 standard or not) and will be analysed using a multilevel logistic regression model. The distribution of potential continuous outcomes such as early-year assessment scores will be assessed before analysing using linear regression. Count data such as the number of attendances for injuries and ingestions will be analysed using a Poisson or negative binomial multilevel regression modelling. A detailed statistical plan will be written and signed off prior to any analysis.

A state transition model using Markov chains will be used to assess the probabilities of moving from one stage marker (states) to another.[25] The transition probabilities (the probability of the various state changes) in our model will be derived from our data and compared between groups.

Bias in the followed-up BB:2–6 sample will be quantified by examining group differences (participants and non-participants) in baseline variables such as age, deprivation, gestational age and education. Surveillance bias in detection of maltreatment during the child's infancy and toddlerhood can be assessed by examining subsequent reporting.[26] The duration between birth and the date of first referral to CSC will be calculated, and group differences will be examined using Cox regression analysis to calculate hazard ratios for referral, together with 95% CIs. Surveillance bias is most likely to occur during the intervention phase, although improved handover to other services at 2 years may lead to higher identification in the following year. Severity of the referral will also be compared between the two groups (an approach used in US trials of NFP to explore surveillance bias).

### Health economics

The economic evaluation will consider costs and consequences of the FNP over the full follow-up period (BB:0–2 and BB:2–6). The current BB:0–2 study reported (1) a within-trial cost utility analysis assessing NHS costs against quality-adjusted life years (QALYs) from the perspective of the mother and (2) a within-trial cost–consequences analysis relating all costs (including those to the social care, education and criminal justice sectors as well as health) against the full range of effects.[12] Cost-and-consequences framework is deemed the most appropriate economic evaluation framework for public health interventions[27] and preferred by National Institute for Health and Care Excellence[28] because it enables capture of equity consideration as well as intersectoral costs and consequences,[29] yet applications are still limited.[27]

The absence of additional data on health-related quality of life within the BB:2-6 study means that it will not be possible to estimate QALYs beyond 24 months

post partum and hence extend the within-trial cost utility analysis. However, the within-trial cost–consequences analysis will be extended from 0–2 to 0–6 years through collection of resource use data from medical and education records (including from the latter, data related to social care usage). Costs will be summarised against the range of outcomes collected within BB:2-6 without aggregation to allow weighing up changes in the various outcomes reported in BB:2-6 against the changes in costs in a consistent and transparent manner.[30] This will contribute to providing more robust and valid medium-term estimates within the extended period.

## ETHICS AND DISSEMINATION
### Legal and ethical considerations
The potential for using routine data in health and social care research has been greatly publicised, and study designs utilising these data are encouraged by funders.[31] There are, however, many inherent challenges in working with secondary-use data, in particular for this project the ethical and legal requirements/responsibilities, which have fundamentally informed this study design.

Although BB:0–2 linked trial data to HES and ONS data via NHS Digital, the governance requirements around the two applications have differed between the two studies not least because of the difference in consent models. Trial data were provided by NHS Digital and ONS after participant consent to prospective collection and for specified purposes limited to the time frame of that study. The current follow-on study uses a dissent model under which we are only able to send trial participant identifiers to ICs for matching to outcome data records if there is no objection received from mothers. This is especially important, as following an opportunity to object to being included in the current study, those women who withdrew from the original Building Blocks will be retained. The study will require all clinical, social and educational data to be held in a data safe haven using encrypted record identifiers and analysis via a securely managed and monitored remote portal. The legal bases for transfer of identifiable data to ICs without explicit consent are as follows; s251 of the 2006 NHS Act 2006 for HES data from NHS Digital, s42(4) of the Statistics and Registration Service Act 2007 through National Institute for Health Research funding for ONS data via NHS Digital and 6 (1) of Schedule 2 of the 1998 Data Protection Act for NPD data.

### Dissemination of findings
The Building Blocks: 2–6 Study will generate policy-relevant findings describing the medium-term impact of FNP on measurements of child maltreatment. The findings will also include other policy-relevant outcomes from the programme such as healthcare use, education attainment and changes in social care use over the 6 years of follow-up. Such medium-term evaluation remains important as some outcomes for the intervention are expected to arise only after the child's second birthday, including maltreatment.

This study will either confirm the largely negative trial findings from BB:0–2 further weakening the justification for FNP Programme continuation or provide a balance to the early measurable outcomes.

In addition to reporting the findings to the funder for this study, the funder for the BB:0–2 trial (DoH Policy Research Programme) will also be informed as well as the FNP National Unit. All local authorities in England will be notified of the results, as (since October 2015) they have responsibility for commissioning public health services for children aged 0–5. Participants will receive a summary of the results, and all reports and publications will be made publicly available in full on the Cardiff University website. The research team has previously convened and met twice with a stakeholder group, including relevant policy leads from each country in the UK delivering FNP (England, Scotland, Northern Ireland). We will stage a similar event to present and discuss the implications for practice and policy of the results of this medium-term follow-up of participants.

In addition to policy and public outputs, academic outputs will include (1) this protocol paper providing visibility of this medium-term follow-up, (2) a methods paper describing the piloting process of the study (including data quality and success of data matching) and (3) main study findings. We aim to disseminate in high-quality, peer-reviewed journals and present in key conferences.

A particular benefit of this study is understanding of, and learning from, the governance challenges. There is potential to use this method for future trials looking at longer-term follow-up. Therefore, this study has the potential to add to the understanding of routine data and data linkage methods in future public health and clinical trials, and these planned publications will provide a basis for the dissemination of the success of these methods.

Finally, publishing protocol papers in medical journals was an important innovation for trials. They convey a number of benefits including transparency about what was intended by researchers and therefore comparison to what was actually reported. While protocols are more commonly published for trials, we consider that the protections afforded are similar for other study types. This may include inhibiting 'data dredging' and post-hoc revisions to original study plans. In our study, which links a trial cohort to routine data, we consider that this is especially important particularly because of the broad range of outcomes that are potentially impacted by this complex home-visiting intervention.

**Author affiliations**
[1]South East Wales Trials Unit, Centre for Trials Research, Cardiff University, Cardiff, UK
[2]Swansea Centre for Health Economics, Swansea University, Swansea, UK
[3]College of Medicine, Swansea University, Swansea, UK
[4]Division of Population Medicine, Cardiff University School of Medicine, Cardiff, UK
[5]Faculty of Life Sciences and Education, University of South Wales, Pontypridd, UK
[6]Marie Curie Palliative Care Research Centre, Cardiff University School of Medicine, Cardiff, UK
[7]Department of Psychology, University of Derby, Derby, UK
[8]School of Healthcare Sciences, Cardiff University, Cardiff, UK

⁹DECIPHer Centre, Cardiff School of Social Sciences, Cardiff University, Cardiff, UK

**Acknowledgements** The South East Wales Trials Unit is funded by Health and Care Research Wales. The work was undertaken with the support of the Centre for the Development and Evaluation of Complex Interventions for Public Health Improvement, a UK Clinical Research Collaboration (UKCRC) Public Health Research Centre of Excellence. Joint funding (MR/K0232331/1) from the British Heart Foundation, Cancer Research UK, Economic and Social Research Council, Medical Research Council, the Welsh Government and the Wellcome Trust, under the auspices of UKCRC, is gratefully acknowledged.

**Contributors** MR is the chief investigator of the study. All authors have contributed to and are responsible for the final design of the study. FLW and GM are responsible for study and data management. RCJ is responsible for statistical planning and for data analysis. DF is responsible for the health economics. All authors read and approved the final manuscript.

**Funding** This project was funded by the National Institute for Health Research Public Health Research (NIHR PHR) Programme (project number 11/3002/11). The views and opinions expressed therein are those of the authors and do not necessarily reflect those of the NIHR PHR Programme or the Department of Health.

**Competing interests** None declared.

**Ethics approval** Ethics approval of the study has been given by the Research Ethics Committee for Wales (14/WA10062), and the transfer and use of identifiable data has been approved by the Health Research Authority Confidentiality Advisory Group (CAG 10-08(b)/2014).

**Provenance and peer review** Not commissioned; externally peer reviewed.

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
