## [Reviewer comments · BMJ Open]

ARTICLE DETAILS

TITLE (PROVISIONAL)	Assessing the medium-term impact of a home-visiting programme on child maltreatment in England: protocol for a routine data linkage study
AUTHORS	Lugg-Widger, Fiona; Cannings John, Rebeca; Channon, Sue; Fitzsimmons, Deborah; Hood, Kerenza; Jones, Kerina; Kemp, Alison; Kenkre, Joyce; Longo, Mirella; McEwan, Kirsten; Moody, Gwenllian; Owen-Jones, Eleri; Sanders, Julia; Segrott, Jeremy; Robling, Michael

VERSION 1 - REVIEW

REVIEWER	Jennifer Fraser The University of Sydney
REVIEW RETURNED	24-Feb-2017

GENERAL COMMENTS	I am not convinced that the power analysis is adequate. Can this be reviewed by a statistician on my behalf for this review. If it is correct, I recommend the manuscript be accepted for publication.
--

REVIEWER	Lynn Kemp Western Sydney University
REVIEW RETURNED	27-Feb-2017

GENERAL COMMENTS	The paper describes the follow on study for the Building Blocks trial. There are a small number of clarifications needed. 1. For the international audience, a description of the UK child protection system would be most helpful - for example, what and who defines a child as 'in need'; what are the services provided by social care (in other countries this may refer to third sector services such as parenting groups etc, but it is clear that this refers to some form of child protection services in this context)?2. There is a small are of lack of clarity with regards to the sample for the BB:2-6. Line 192 defines eligible participants as those exiting the BB:0-2 study. This would imply that those eligible are limited to those who completed the 0-2 study, however, subsequent text suggests that those eligible for BB:2-6 are all those who entered the original trial. Can the authors please clarify this: in particular, is participant withdrawal from BB:0-2 also interpreted as withdrawal from follow-up. If not, what are the ethical issues of including data and conducting follow-up for those who withdrew from the study?3. Line 332 refers to key subgroups: can the authors please detail these.4. The primary source of data eligibility for a number of the key data items is public school data. Is completeness of data potentially impacted by non-participation by children in the public school
--

	system, particularly in the younger years? A description of the publically funded school system (including participation rates in prior to school programs) would be helpful for the international audience. Minor editorial issues: Line 361 is unclear (probable missing word). Lines 337-9 sentence has an awkward construction: rewording needed.
--	---

REVIEWER	Howard Dubowitz University of Maryland School of Medicine USA
REVIEW RETURNED	06-Mar-2017

GENERAL COMMENTS	This paper provides a well written detailed description of the Methods for the proposed study, but before the study has started. If the journal publishes such articles, given that the topic is of major public health import, this paper should be of interest. I am not familiar however with seeing such articles in the peer reviewed literature. Rather, I would see this being a supplement to a paper presenting the study's findings.
--

VERSION 1 – AUTHOR RESPONSE

Reviewer 1:

1. I am not convinced that the power analysis is adequate. Can this be reviewed by a statistician on my behalf for this review. If it is correct, I recommend the manuscript be accepted for publication.
RESPONSE: Although the reviewer's comment was not specifically directed toward the authors to address we have added some additional text (line 342-345) to clarify the direction and size of expected effect. We have also provided some more detail of the planned secondary analysis (line 364-372). As the protocol paper describes the follow-up of an existing cohort of therefore fixed size we have described what is possible given the data set we are likely to have available, including for example accounting for loss to follow-up through failure in tracking and linkage. We would be happy to respond to any further statistical review requested.

Reviewer 2:

The paper describes the follow on study for the Building Blocks trial. There are a small number of clarifications needed.

1. For the international audience, a description of the UK child protection system would be most helpful - for example, what and who defines a child as 'in need'; what are the services provided by social care (in other countries this may refer to third sector services such as parenting groups etc., but it is clear that this refers to some form of child protection services in this context)?

RESPONSE: We agree that this would benefit the international audience of the journal and have added a paragraph (lines 77-85) to describe the definition of child in need as well as examples of services provided by local authorities.

2. There is a small are of lack of clarity with regards to the sample for the BB:2-6. Line 192 defines eligible participants as those exiting the BB:0-2 study. This would imply that those eligible are limited to those who completed the 0-2 study, however, subsequent text suggests that those eligible for BB:2-6 are all those who entered the original trial. Can the authors please clarify this: in particular, is participant withdrawal from BB:0-2 also interpreted as withdrawal from follow-up. If not, what are the ethical issues of including data and conducting follow-up for those who withdrew from the study?

RESPONSE: We have added two clarifying statements (lines 218-220 and 420-422). All participants

who were eligible to continue to the end of the trial have been given the opportunity to continue or opt-out of this follow-on study (i.e. elective withdrawals are included). This excludes those who withdrew electively with the option of no further use of their data.

3. Line 332 refers to key subgroups: can the authors please detail these.

RESPONSE: We have added the key subgroups, please see line 359-360.

4. The primary source of data eligibility for a number of the key data items is public school data. Is completeness of data potentially impacted by non-participation by children in the public school system, particularly in the younger years? A description of the publically funded school system (including participation rates in prior to school programs) would be helpful for the international audience.

RESPONSE: We agree that this would be a useful addition to the manuscript and have added in data coverage for School Census as well as the earlier years. This has been added in to lines 190-203 and 214-216.

5. Minor editorial issues: Line 361 is unclear (probable missing word). Lines 337-9 sentence has an awkward construction: rewording needed.

RESPONSE: This has been addressed (now line 395)

Reviewer: 3

1. This paper provides a well written detailed description of the Methods for the proposed study, but before the study has started. If the journal publishes such articles, given that the topic is of major public health import, this paper should be of interest. I am not familiar however with seeing such articles in the peer reviewed literature. Rather, I would see this being a supplement to a paper presenting the study's findings.

RESPONSE: Protocol papers published in medical journals were an important innovation for trials. They convey a number of benefits including transparency about what was intended by researchers and therefore comparison to what was actually reported (ref: <http://old.biomedcentral.com/authors/protocols>).

While protocols are more commonly published for trials, we consider that the protections afforded are similar for other study types. This may include inhibiting 'data dredging' and post-hoc revisions to original study plans. In our study which links a trial cohort to routine data we consider that this is especially important, particularly because of the broad range of outcomes that are potentially impacted by this complex home visiting intervention.

As the reviewer has asked this question we have also included a short paragraph on this matter at the end of the manuscript (line 458-465).

We trust that we have adequately addressed both the editorial comments and those of the three reviewers. We look forward to hearing your response to our re-submission.

VERSION 2 – REVIEW

REVIEWER	Jennifer Fraser University of Sydney, Australia
REVIEW RETURNED	25-Apr-2017

GENERAL COMMENTS	I recommended accept previously. The revisions completed are acceptable. I think the protocol is ready for publication.
---

REVIEWER	Lynn Kemp Western Sydney University Australia
-----------------	--

REVIEW RETURNED	19-Apr-2017
-------------

GENERAL COMMENTS	The authors have adequately addressed all the issues raised by the reviewers. In particular the context for the research and outcomes measured are clearer for the international audience.
--